# Socio-Economic and Demographic Factors Associated with Knowledge and Attitude of HIV/AIDS among Women Aged 15–49 Years Old in Indonesia

**DOI:** 10.3390/healthcare10081545

**Published:** 2022-08-15

**Authors:** Feny Deya Virdausi, Ferry Efendi, Tiyas Kusumaningrum, Qorinah Estiningtyas Sakilah Adnani, Lisa McKenna, Kadar Ramadhan, Ika Adelia Susanti

**Affiliations:** 1Faculty of Nursing, Universitas Airlangga, Surabaya 60115, Indonesia; 2Faculty of Medicine, Universitas Padjadjaran, Bandung 45363, Indonesia; 3School of Nursing and Midwifery, La Trobe University, Melbourne 3086, Australia; 4Department of Midwifery, Poltekkes Kemenkes Palu, Palu 94145, Indonesia

**Keywords:** AIDS, attitude, demographic factors, demographic health, HIV, knowledge, socio-economic, survey, women

## Abstract

Women’s susceptibility to HIV/AIDS infection is related to socio-economic and demographic factors. This study sought to analyze socio-economic and demographic factors related to knowledge and attitude of HIV/AIDS among women aged 15–49 years old in Indonesia. We conducted a secondary data analysis using the 2017 Indonesian Demographic and Health Survey (IDHS). Among 49,627 women, our study analyzed 25,895 women aged 15–49 years familiar with HIV terminology. Multiple logistic regression was utilized to analyze associations between socio-economic and demographic factors with knowledge and attitudes toward HIV/AIDS. Women’s age, education level, wealth quintile, residential area and region, access to information, owning cell phones and autonomy were significantly associated with positive knowledge and attitudes toward HIV/AIDS. These findings revealed that several demographical and social factors contribute to knowledge and attitudes toward HIV/AIDS among women aged 15–49 years in Indonesia.

## 1. Introduction

Human Immunodeficiency Virus/Acquired Immune Deficiency Syndrome (HIV/AIDS) is still a significant global burden [1,2]. Even though substantial efforts have been made to reduce the HIV/AIDS infection rate and its prevalence, in 2019, more than 35 million people were living with HIV/AIDS (PLWHA); the majority reside in Sub-Saharan Africa [1,3,4]. The Southeast Asia region ranks second globally, with many PLWHA. Data from the United Nations (UN) revealed that approximately 19 million people do not know their HIV/AIDS status [5]. The World Health Organization (WHO) stated that 78% of new infections are in the Pacific region. A total of 5.1 million people in the Asia and Pacific Region are infected with HIV, including approximately 1.80 million adult women and 1.60 million adult men. As many as 3.8 million are infected with HIV/AIDS in Southeast Asia [6]. In low and middle-income countries, the prevalence of HIV/AIDS among women is three times higher than for men [7].

In Indonesia, in 2019, the number of HIV/AIDS cases reported was 50,282 [8]. In Indonesia, women are estimated to be four times more likely to be infected than men. Women are considered a vulnerable group infected with HIV/AIDS due to reproductive and genital anatomical structures facilitating the transmission of HIV through sexual intercourse [9]. Women’s vulnerability is formed by several factors, including socio-cultural, economic and biological factors. The interconnection between gender inequality, migration, barriers to accessing health services and low levels of education increase women’s vulnerability to contracting HIV/AIDS. In addition, many women contract HIV/AIDS from their husbands/partners who have unsafe sexual behavior and use drugs [5].

Lack of knowledge may lead to negative attitudes towards PLWHA. It could contribute to insufficient practice for prevention, treatment, and the risk of transmission among the community. Knowledge about HIV/AIDS and its transmission can be mainly achieved by increasing acceptance towards PLWHA and making people more caring [10,11]. Several studies have demonstrated that people tend to have negative attitudes and report not buying vegetables from PLWHA-status sellers [12,13,14]. These PLWHA have physical, economic, social and psychological consequences as HIV-infected patients are reportedly rejected by healthcare services [14], experience poor quality treatment, are refused when applying for jobs due to HIV/AIDS status, and experience forced early resignation from their employment [12,13]. 

Knowledge and attitudes may affect behaviour toward HIV/AIDS [15]. Sufficient knowledge and positive attitudes concerning HIV/AIDS provide empirical evidence for policymakers and stakeholders with which to design and implement appropriate prevention mechanisms [16,17]. Previous studies in East Africa and Vietnam revealed that lack of knowledge among women can be attributed to the negative attitudes towards PLWHA [15,18]. Some previous studies concluded that good knowledge and a positive attitude are important indicators in the prevention of HIV transmission [19,20,21]. One study among women of childbearing age in South Sudan showed that women living in urban areas had better knowledge compared to women living in rural areas [22]. Several studies have been conducted concerning HIV/AIDS prevention [23,24], while scarce literature addresses socio-economic and demographic factors associated with knowledge and attitudes toward HIV/AIDS in Indonesia. Knowledge and attitudes concerning HIV/AIDS among Indonesian women remain critical concerns as some studies revealed insufficient knowledge and negative attitudes [25,26,27]. Further, recent studies regarding the knowledge and attitudes of HIV/AIDS in Indonesia reemphasized that behavioral issues remain a challenge [9,28]. 

This study can aid in establishing knowledge, provide a basis for further research in HIV/AIDS and assist the government in strengthening and modifying the program to approach the HIV/AIDS issue. Possessing good knowledge and a positive attitude in relation to HIV/AIDS is very important for avoiding HIV transmission and to end the discrimination among PLWHA. There is increasing concern that some women of reproductive age lack accurate and complete information on how to prevent HIV/AIDS transmission. In line with these concerns, the objective of our study was to assess socio-economic and demographic factors related to knowledge and attitudes toward HIV/AIDS among women aged 15–49 years in Indonesia.

## 2. Methods

### 2.1. Design and Data Source

We conducted a secondary data analysis utilizing the most recent data from the 2017 Indonesia Demographic Health and Survey (IDHS). This study was part of an international DHS program that ensured ethical standards, including confidentiality, anonymity, and informed consent. IDHS ethical clearance was obtained from the Inner City Fund (ICF) International (ethical approval number 45 CFR 46). For this study, permission to use the data was obtained from ICF International. Additionally, ethical approval was obtained before the survey was conducted, and all participants provided written informed consent. All participation in this study was voluntary and subjects were able to withdraw their participation at anytime from the study. 

This study used women’s questionnaire topics which included questions that assessed women’s knowledge of HIV and other sexually transmitted infections, the sources of their knowledge about HIV, knowledge about ways to avoid contracting HIV, HIV testing, stigma and discrimination, and high-risk sexual behaviour [29]. A series of questions on questionnaires about HIV/AIDS was required to be answered by women related to the (DHS) standard. The Model Questionnaires of the DHS Program emphasize basic indicators and several modules [30]. The data quality issue is a serious concern and sustained attention from DHS implementers to improve the validity and reliability of questionnaire is evident [31]. Continual improvement of the methodology including the questions for each questionnaire is performed collaboratively among various stakeholders. The questionnaire is open and available to the public at the DHS website, which can be accessed using the following link: https://dhsprogram.com/pubs/pdf/DHSQ8/DHS8_Womans_QRE_EN_8Apr2022_DHSQ8.pdf (accessed on 9 April 2022). 

There were sections on knowledge and attitudes toward HIV/AIDS in the questionnaires. Questions on knowledge of HIV/AIDS included reducing risk, and concerned aspects such as always using a condom during sexual intercourse, only having one partner, getting HIV through mosquito bites or by sharing food with PLWHA, that people who appear healthy can have HIV, can contract HIV through supernatural powers, getting HIV through unsterilized needles, and that HIV can be transmitted during pregnancy, childbirth and breastfeeding. Questions on attitudes toward HIV/AIDS included wanting to keep HIV infection in the family a secret, willingness to care for families with AIDS, whether children with HIV are allowed to go to school with children who are not HIV positive and whether they would buy vegetables from a seller who has HIV.

The cross-sectional study represented 1970 census blocks of urban and rural areas covering 49,250 households across 34 provinces in Indonesia. This survey was conducted in several steps, including a pretest (July–August 2016), training of field staff (July 2017), and fieldwork (24 July–30 September 2017). A two-stage stratified cluster sampling method was employed to recruit the respondents in this study. First, several census blocks were selected by performing systematic sampling of the proportional size. Second, 25 ordinary households were chosen from the list via systematic sampling. With these data, the inclusion criteria for our study were as follows: aged 15–49 years, those who were interviewed during the survey and participants who completed all of the questions including those on HIV/AIDS issues. We excluded women who did not answered and complete all of the questions of the survey. Women’s weight was obtained and individual recording of data during the analysis was applied. Among 49,627 women, our study analyzed 25,895 women aged 15–49 years based on the inclusion criteria. This study was a representative of national study involving women over all provinces in Indonesia. 

### 2.2. Variables

The dependent variables were knowledge and attitudes toward HIV/AIDS. In this study, knowledge on HIV/AIDS was categorized into the following two groups: poor and good, while attitudes on HIV/AIDS were categorized into positive and negative. Knowledge was categorized as poor if respondents only answered less than five correctly, and categorized as good if respondents answered ≥ 5 correctly. Attitudes were divided into positive if respondents answered ≥3 questions correctly and negative if they only answered 1–2 questions correctly. In this study, the independent variables related to socioeconomic and demographic factors, including age, education level, occupation, head of household, wealth quintile, area of residence, region of residence, access to information, mobile phone, autonomy, and women’s attitudes against wife-beating. The women’s age was divided into seven categories (15–19, 20–24, 25–29, 30–34, 35–39, 40–44, 45–49 years old). Women’ education levels were grouped into no school, primary, secondary, and higher education. Women’ employment status was divided into two categories of not working and working. The head of the household was divided into two categories of women and men. Wealth quintile was grouped into the following five categories: poorest, poorer, middle, richer, and richest, and was scored based on wealth criteria on the DHS report [32]. Place of residence was separated into rural and urban areas, while the provinces of Indonesia were defined as East, Middle and West. Access to information was categorized into never, less than once, and more than once per week. Mobile phones were divided into yes and no, while women’s autonomy was grouped into high and low. Women’s attitudes against wife-beating was divided into two categories of agree and disagree. All categories’ variables were determined based on the DHS report that adjusted to the minimum sample to meet the statistical interpretation [33].

### 2.3. Statistical Analysis

Descriptive statistics were used to describe the characteristics of the respondents. We used the chi-square test to assess associations between socio-economic and demographic factors and knowledge and attitudes toward HIV/AIDS. A multiple logistic regression analysis was performed, which presented an Odds Ratio (OR) and a 95% Confidence Interval (CI) to measure the variables’ associations. The statistical significance was identified if a *p*-value was under 0.05, which was considered for results to enter the multivariate analysis. All statistical analyses were carried out using Stata 16.

## 3. Results

### 3.1. Characteristics of the Respondents

Table 1 presents the distribution of knowledge and attitudes toward HIV/AIDS according to sociodemographic and demographic factors among women aged 15–49 years old in Indonesia. A total of 25,895 women who had complete HIV/AIDS data were included in the analysis, and 88.74% of women had a high level of knowledge, while 60.28% had negative attitudes toward PLWHA. Nearly one quarter (20.64%) were 35–39 years old. More than half (59.15%) of the respondents had received secondary education and were working (61.27%). Close to all of the respondents’ husbands were heads of the household (93.08%), while one quarter (25.15%) belonged to the wealthiest families. More than half (54.41%) resided in urban areas, while more than three quarters (84.51%) of the respondents lived in the West of Indonesia. More than half reported never accessing the Internet (55.76%), newspaper (58.93%), and radio (58.64%). More than three-quarters of the respondents’ accessed information more than once per week from television (88.10%), had a mobile phone (81.89%), had high autonomy (85.97%), and disagreed with wife-beating (82.49%). Further information about the respondents’ characteristics is presented in Table 1.

### 3.2. Bivariate Analysis

In the bivariate analysis, an age of 15–19 years, lack of formal education, belonging to the poorest index, living in rural and West Indonesia, having no access to media, having no mobile phone, low autonomy and agreeing with wife-beating were associated with knowledge toward HIV/AIDS (Table 2). Similarly, an age of 15–19 years, having no formal education, belonging to the poorest index, living in rural and the West of Indonesia, having no access to the Internet, newspaper, radio, having no mobile phone, and agreeing with wife-beating were associated with attitudes toward HIV/AIDS (Table 3). No significant association was detected between an age of 20–49 years, occupation, and head of household.

### 3.3. Multiple Logistic Regression Analysis

The final multiple logistic regression models in Table 4 were adjusted for association across variables. People aged 45–49 years old were 181% more likely to have good knowledge of HIV/AIDS compared to women aged 15–19 [1.81(1.26–2.60)]. Women with higher education levels were 6 times more likely to have good knowledge compared to women who received no education at all [6.32(3.59–11.11)]. Women in the richest wealth index were 167% more likely to have good knowledge of HIV/AIDS compared to women with the poorest wealth index [1.67(1.32–2.10)]. Women who lived in urban areas and East of Indonesia Province were 137% [1.37(1.22–1.54)] and 128% [1.28(1.00–1.68)] more likely to have good knowledge compared to those in rural areas and the middle Indonesia Provinces, respectively. Women who accessed information from the internet ≥1 per week were 174% more likely to have good knowledge compared to women who never access the internet [1.74(1.50–2.03)]. Women who were exposed to a newspaper ≥1 per week were as much as 132% more likely to have good knowledge than those never exposed to newspaper [1.32(1.04–1.67)]. This study found that women who access the information through radio ≥1 per week have a 131% [1.31(1.11–1.56)] propensity to acquire good knowledge compared to women who never access radio. Women who access information through a mobile phone were 126% [(1.26 (1.12–1.44)] more likely to have good knowledge compared to those who do not access from a mobile phone. Women with high autonomy were 123% more likely to have good knowledge compared to those with low autonomy [1.23(1.05–1.44)].

There was no association observed in terms of access to television, newspaper and autonomy with positive attitudes toward HIV/AIDS. 

## 4. Discussion

In this study, we found socio-economic and demographic factors were associated with knowledge and attitudes of HIV/AIDS among women aged 15–49 years. Among the representative sample in this study, more than three quarters (88.74%) had a high level of knowledge, while more than half (60.28%) of women had negative attitudes toward PLWHA, revealing the tendencies of women in understanding information on HIV/AIDS. Even though the study population was significantly knowledgeable about HIV/AIDS, negative attitudes towards PLWHA showed that accepting PLWHA still requires reasonable efforts and resources. Our study highlights the need for public and/or specific group awareness about HIV/AIDS, as suggested in previous studies [34,35,36]. Negative attitudes toward PLWHA may lead to persistent discrimination and their persistent rejection by community members [37,38,39]. Our study indicated that a high level of knowledge about HIV/AIDS does not translate to more positive attitudes and revealed the critical social barrier for PLWHA in Indonesia.

Our findings revealed that women’s age played a vital role in possessing a high level of knowledge, while age was not associated with attitudes toward HIV/AIDS. Consistent with previous studies [20,40,41], age was associated with the person’s opportunity to gather additional and appropriate information considered necessary for daily life. Mature age can be attributed to greater exposure to sexual health education, such as training related to sexual health and HIV/AIDS. One explanation why age was not associated with attitudes might be because age is related to someone’s experience during their lifetime, while attitude represents a complex processes within human perception that can change constantly depending on specific situations [42]. 

Our study confirmed the findings of previous studies indicating that higher education was associated with knowledge and attitudes toward HIV/AIDS among women of reproductive age [20,43,44]. This may be due to educational attainment acts which have made information more easily accessible and better promoted the reception of such knowledge [18]. Further, appropriate resources might foster better knowledge and attitudes toward PLWHA, which can assist in the problem of HIV/AIDS infection, treatment, and transmission [18,45,46].

Further, our study found that socio-economic and demographic factors, including residing in urban areas, living in the West of Indonesia, having access to mass media, and having mobile phones were associated with knowledge and attitudes on HIV/AIDS. These results are similar to previous studies conducted in other countries such as India, Bangladesh, and Pakistan [34,47,48]. Our findings indicate an urgent need to target women from urban areas and who are exposed to mass media through appropriate campaigns [49]. Related to the current context of infectious diseases, the easy availability and accessibility of health information online can improve patient knowledge and practice related to HIV/AIDS [50,51]. A similar study also found that most people have a good level of knowledge and understand of preventative actions related to infectious disease such as COVID-19 [52]. In the Indonesian setting, robust policies and strategic programs have utilized the ABCDE (Abstinence, Be faithful, Condom, Do not use drugs, Education) campaigns to reduce the risk of contracting HIV in the East of Indonesia. A series of efforts has been made through promotional activities, counselling, and voluntary testing and treatment; however, HIV/AIDS cases remain high in East of Indonesia, especially in the Papua region [8]. Practical policies supplemented with local insights may need to be tested at the provincial level to understand how to achieve a better outcome. 

Moreover, our study revealed that those women who were well-educated and with a more affluent wealth index were more likely to show good knowledge and positive attitudes towards HIV/AIDS. This finding aligns with another study conducted in South Sudan where wealth quintiles had a significant relationship with comprehensive knowledge and positive attitudes towards people with HIV/AIDS [22]. The ability of women to actively empower themselves with knowledge on HIV/AIDS relates to the personal awareness and comprehensive understanding of women about HIV/AIDS. These findings are also consistent with other studies conducted in Ethiopia and Pakistan [47,52] which suggested that women’s autonomy is vital to address the effect of HIV/AIDS on women’s health. Further, our findings showed that women who disagreed with wife-beating were more likely to have positive attitudes towards HIV/AIDS [47,52,53].

This study has certain strengths that should be highlighted. Our paper is the first survey report on socio-economic and demographic factors associated with knowledge and attitudes of HIV/AIDS among women aged 15–49 years old in Indonesia using well-recommended global tools. All data utilized in the analysis were weighted to reflect the statistical interpretation with rigorous methodology and techniques. A significant limitation of our study related to the completeness of data available from the DHS website [33], which intended to capture demographic and health indicators in Indonesia. The limitation of this study was found to be the study design which was cross-sectional; therefore, we cannot infer the causality here. Due to the use of secondary data, we also had no control over the confounding factors and indicators. Despite these limitations, the findings are important for the formulation of more effective policies concerning knowledge and attitudes toward HIV/AIDS. 

## 5. Conclusions

This study analyzed socio-economic and demographic factors associated with knowledge and attitudes toward HIV/AIDS among women aged 15–49 years old in Indonesia. In order to combat HIV/AIDS in Indonesia, issues of knowledge and attitudes toward HIV/AIDS still must be addressed. In general, women with a higher level of education, higher wealth status, living in an urban area, residing in West Indonesia and having access to the Internet, radio, and mobile phones had significantly better levels of knowledge and positive attitudes towards HIV/AIDS. Our findings indicate that HIV/AIDS knowledge and attitudes related to personal background, place, and mode of media contributed to a high level of knowledge and positive attitudes towards HIV/AIDS. Appropriate health education programs are recommended as the key to increasing the level of comprehensive knowledge and attitudes related to HIV/AIDS among women. A health education campaign should be launched based on sociodemographic considerations by working with the local governments and relevant stakeholders. 

## Figures and Tables

**Table 1 healthcare-10-01545-t001:** Characteristics of women aged 15–49 years in Indonesia.

Characteristics	N	%
**Age (Years)**		
15–19	443	1.71
20–24	2561	9.89
25–29	4372	16.88
30–34	5106	19.72
35–39	5345	20.64
40–44	4425	17.09
45–49	3643	14.07
**Education level**		
No school	112	0.43
Primary	6398	24.71
Secondary	15,316	59.15
Higher	4069	15.71
**Occupation**		
Not working	10,029	38.73
Working	15,866	61.27
**Head of household**		
Men	24,104	93.08
Women	1791	6.92
**Wealth index**		
Poorest	2982	11.52
Poorer	4685	18.09
Middle	5507	21.27
Richer	6209	23.98
Richest	6512	25.15
**Resident**		
Rural	11,806	45.59
Urban	14,089	54.41
**Province**		
West of Indonesia	21,884	84.51
Middle of Indonesia	3485	13.46
East of Indonesia	526	2.03
**Access to Internet**		
Never	14,439	55.76
<1 per week	648	2.50
≥1 per week	10,808	41.74
**Access to television**		
Never	515	1.99
<1 per week	2567	9.91
≥1 per week	22,813	88.10
**Exposure to newspaper**		
Never	15,259	58.93
<1 per week	7991	30.86
≥1 per week	2645	10.21
**Access to radio**		
Never	15,184	58.64
<1 per week	7035	27.17
≥1 per week	3676	14.20
**Mobile phone**		
No	4690	18.11
Yes	21,205	
**Autonomy**		
Low	3632	14.03
High	22,263	85.97
**Women’s attitudes towards wife-beating**		
Agree	4535	17.51
Disagree	21,360	82.49
**Knowledge toward HIV/AIDS**		
Poor	2915	11.26
Good	22,980	88.74
**Attitudes toward PLWHA**		
Negative	15,610	60.28
Positive	10,285	39.72

HIV/AIDS: Human Immunodeficiency Virus/Acquired Immune Deficiency Syndrome. PLWHA: People Living with HIV/AIDS.

**Table 2 healthcare-10-01545-t002:** Socioeconomic and demographic information and knowledge toward HIV/AIDS among women aged 15–49 years old in Indonesia.

Characteristics	Knowledge				
	Poor		Good		χ^2^
	*n* (2915)	%	*n* (22,980)	%	
**Age (years)**					
15–19	82	18.55	360	81.45	39.01 ***
20–24	308	12.03	2253	87.97	
25–29	489	11.19	3883	88.81	
30–34	540	10.57	4566	89.43	
35–39	542	10.14	4803	89.86	
40–44	490	11.08	3935	88.92	
45–49	464	12.72	3180	87.28	
**Education level**					
No school	37	33.05	75	66.95	891.11 ***
Primary	1318	20.60	5080	79.40	
Secondary	1453	9.49	13,863	90.51	
Higher	107	2.63	3962	97.37	
**Occupation**					
Not working	1185	11.81	8844	88.19	4.70
Working	1730	10.90	14,136	89.10	
**Head of household**					
Men	2726	11.31	21,378	88.69	0.87
Women	189	10.56	1602	89.44	
**Wealth index**					
Poorest	596	19.98	2386	80.02	679.12 ***
Poorer	779	16.63	3906	83.37	
Middle	715	12.99	4792	87.01	
Richer	546	8.79	5663	91.21	
Richest	279	4.28	6233	95.72	
**Resident**					
Rural	1824	15.45	9983	84.55	353.28 ***
Urban	1091	7.74	12,997	92.26	
**Province**					
West of Indonesia	2382	10.89	19,502	89.11	18.00 ***
Middle of Indonesia	464	13.30	3022	86.70	
East of Indonesia	69	13.13	456	86.87	
**Access to internet**					
Never	2283	15.81	12,156	84.19	651.30 ***
<1 per week	76	11.66	573	88.34	
≥1 per week	556	5.15	10,251	94.85	
**Access to television**					
Never	83	16.08	432	83.92	15.81 **
<1 per week	320	12.46	2247	87.54	
≥1 per week	2512	11.01	20,301	88.99	
**Exposure to newspaper**					
Never	2075	13.6	13,184	86.40	236.43 ***
<1 per week	732	9.16	7259	90.84	
≥1 per week	108	4.08	2537	95.92	
**Access to radio**					
Never	1937	12.76	13,247	87.24	83.40 ***
<1 per week	684	9.72	6351	90.28	
≥1 per week	294	7.99	3382	92.01	
**Mobile phone**					
No	918	19.58	3771	80.42	367.78 ***
Yes	1997	9.42	19,209	90.58	
**Autonomy**					
Low	508	13.97	3124	86.03	28.90 ***
High	2407	10.81	19,856	89.19	
**Women’s attitudes towards wife-beating**					
Agree	631	13.90	3905	86.10	35.71 ***
Disagree	2284	10.69	19,075	89.31	

** χ^2^ < 0.01. *** χ^2^ < 0.001.

**Table 3 healthcare-10-01545-t003:** Socioeconomic and demographic information and attitudes toward HIV/AIDS among women aged 15–49 years old in Indonesia.

Characteristics	Attitudes				
	Negative		Positive		χ^2^
	*n* (15,610)	%	*n* (10,285)	%	
**Age (years)**					
15–19	274	62.00	168	38.00	59.36 ***
20–24	1614	63.01	948	36.99	
25–29	2585	59.13	1787	40.87	
30–34	2937	57.53	2168	42.47	
35–39	3132	58.59	2213	41.41	
40–44	2713	61.32	1712	38.68	
45–49	2355	64.62	1289	35.38	
**Education level**					
No school	87	77.48	25	22.52	294.53 ***
Primary	4326	67.62	2071	32.38	
Secondary	9136	59.65	6180	40.35	
Higher	2061	50.64	2009	49.36	
**Occupation**					
Not working	6112	60.95	3916	39.05	2.81
Working	9498	59.86	6369	40.14	
**Head of household**					
Men	14,514	60.21	9590	39.79	0.62
Women	1096	61.19	695	38.81	
**Wealth index**					
Poorest	2110	70.76	872	29.24	256.14 ***
Poorer	2993	63.87	1693	36.13	
Middle	3371	61.22	2135	38.78	
Richer	3605	58.06	2604	41.94	
Richest	3531	54.22	2981	45.78	
**Resident**					
Rural	7579	64.19	4228	35.81	128.11 ***
Urban	8031	57.01	6057	42.99	
**Province**					
West of Indonesia	12,846	58.70	9039	41.30	140.80 ***
Middle of Indonesia	2423	69.53	1062	30.47	
East of Indonesia	341	64.92	184	35.08	
**Access to internet**					
Never	9413	65.19	5026	34.81	317.08 ***
<1 per week	396	61.13	252	38.87	
≥1 per week	5801	53.67	5007	46.33	
**Access to television**					
Never	311	60.32	204	39.68	0.07
<1 per week	1541	60.04	1026	39.96	
≥1 per week	13,758	60.31	9054	39.69	
**Exposure to newspaper**					
Never	9482	62.14	5777	37.86	84.35 ***
<1 per week	4738	59.29	3253	40.71	
≥1 per week	1390	52.56	1255	47.44	
**Access to radio**					
Never	9286	61.16	5898	38.84	28.74 ***
<1 per week	4259	60.55	2776	39.45	
≥1 per week	2065	56.17	1611	43.83	
**Mobile phone**					
No	3214	68.55	1475	31.45	151.14 ***
Yes	12,396	58.46	8810	41.54	
**Autonomy**					
Low	2207	60.76	1425	39.24	0.37
High	13,403	60.21	8860	39.79	
**Women’s attitudes towards wife-beating**					
Agree	2909	64.15	1626	35.85	31.73 ***
Disagree	12,701	59.46	8659	40.54	

*** χ^2^ < 0.001.

**Table 4 healthcare-10-01545-t004:** Multiple logistic regression analysis of socioeconomic and demographic and knowledge-attitudes toward HIV/AIDS among women aged 15–49 years old in Indonesia.

Variable	Good Knowledge			Positive Attitudes		
	OR	95% CI		OR	95% CI	
		Lower	Upper		Lower	Upper
**Age (years)**						
15–19	1.00			1.00		
20–24	1.40	0.97	2.01	0.87	0.65	1.16
25–29	1.47 *	1.04	2.09	1.00	0.75	1.33
30–34	1.77 **	1.25	2.51	1.11	0.84	1.46
35–39	2.12 ***	1.49	3.00	1.10	0.83	1.46
40–44	2.04 ***	1.44	2.90	1.02	0.77	1.36
45–49	1.81 **	1.26	2.60	0.90	0.68	1.20
**Education level**						
No school	1.00			1.00		
Primary	1.54	0.92	2.57	1.35	0.75	2.41
Secondary	2.86 ***	1.71	4.78	1.64	0.922	2.93
Higher	6.32 ***	3.59	11.11	2.05 *	1.14	3.68
**Wealth index**						
Poorest	1.00			1.00		
Poorer	0.98	0.83	1.16	1.21 **	1.06	1.38
Middle	1.08	0.91	1.28	1.23 **	1.08	1.40
Richer	1.29 **	1.07	1.56	1.26 **	1.10	1.44
Richest	1.67 ***	1.32	2.10	1.22 **	1.05	1.41
**Resident**						
Rural	1.00			1.00		
Urban	1.37 ***	1.22	1.54	1.11 *	1.03	1.21
**Province**						
West of Indonesia	1.27 ***	1.12	1.45	1.59 ***	1.46	1.73
Middle of Indonesia	1.00			1.00		
East of Indonesia	1.28 *	1.00	1.68	1.32 **	1.12	1.57
**Access to Internet**						
Never	1.00			1.00		
<1 per week	1.17	0.87	1.57	1.13	0.92	1.38
≥1 per week	1.74 ***	1.50	2.03	1.26 **	1.15	1.37
**Access to television**						
Never	1.00			1.00		
<1 per week	0.85	0.59	1.22	0.79	0.61	1.03
≥1 per week	0.98	0.70	1.37	0.78	0.62	1.00
**Exposure to newspaper**						
Never	1.00			1.00		
<1 per week	1.07	0.95	1.20	1.02	0.94	1.11
≥1 per week	1.32 *	1.04	1.67	1.07	0.95	1.20
**Access to radio**						
Never	1.00			1.00		
<1 per week	1.10	0.97	1.24	0.96	0.89	1.05
≥1 per week	1.31 **	1.11	1.56	1.13 *	1.03	1.24
**Mobile phone**						
No	1.00			1.00		
Yes	1.26 ***	1.12	1.44	1.17 **	1.06	1.29
**Autonomy**						
Low	1.00			1.00		
High	1.23 **	1.05	1.44	0.99	0.89	1.09
**Women’s attitudes towards wife-beating**						
Agree	1.00			1.00		
Disagree	1.08	0.95	1.22	1.10 *	1.00	1.19

* *p*-value < 0.05. ** *p*-value < 0.01. *** *p*-value < 0.001.

## Data Availability

Data are available from https://dhsprogram.com/data/available-datasets.cfm (accessed on 4 January 2022) by applying through the DHS program via the website. The authors had no special access privileges to these data.

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
