# Peer review of "Socio-Economic and Demographic Factors Associated with Knowledge and Attitude of HIV/AIDS among Women Aged 15–49 Years Old in Indonesia"

_healthcare, 2022, doi:10.3390/healthcare10081545_

Round 1
Reviewer 1 Report
Thank you for allowing me to review this manuscript. This paper entitled "Socioeconomic and demographic factors associated with knowledge and attitude towards HIV/AIDS among women aged 15-49 in Indonesia", aims to assess socioeconomic and demographic factors related to knowledge and attitude towards HIV/AIDS. HIV/AIDS. AIDS among women aged 15-49 in Indonesia.
It is an interesting and highly relevant article today, although it has several limitations that make it suitable for publication in this journal. These limitations are detailed below:
- The citation regulations required by the journal are not strictly followed, since there are errors in the citations. It is necessary to review the citations and adapt them to the regulations indicated by the journal. For example, in the introduction: [1],[2].
- I would recommend that the authors justify the novelty and relevance of the study being carried out in more detail in the Introduction.
- In the material and methods section, it would be necessary to specify some aspects in more detail. I would recommend giving more details about the collection of the sample. How long did the data collection take? Is the sample representative? It is also necessary to specify in detail the inclusion and exclusion criteria applied among the participants.
- The material and methods section does not reflect different important ethical considerations either. It would be necessary to indicate if the authorization of an ethics committee was obtained for data collection. It also does not reflect whether the participation of the patients was voluntary or if they were offered a document with the information and informed consent.
- The results are clear and precise, including the necessary elements and offering greater rigor to the article. However, in the tables the acronyms included in them are not specified at the foot of the table.
- The conclusions are specific and very appropriate. In addition, they point out the implication of the subject in clinical practice and today.
- The bibliographic references do not comply with the regulations required by the journal. I would recommend reviewing and adapting
Author Response
Response to Reviewer 1
Comments
Point 1: It is an interesting and highly relevant article today, although it has several limitations that make it suitable for publication in this journal. These limitations are detailed below:
The citation regulations required by the journal are not strictly followed, since there are errors in the citations. It is necessary to review the citations and adapt them to the regulations indicated by the journal. For example, in the introduction: [1],[2].
Response 1: Thank you for your feedback. We have reviewed the citations style and revise it.
Point 2: I would recommend that the authors justify the novelty and relevance of the study being carried out in more detail in the Introduction.
Response 2: Many thanks, we have added more justification on the introduction section.
Point 3: In the material and methods section, it would be necessary to specify some aspects in more detail. I would recommend giving more details about the collection of the sample. How long did the data collection take? Is the sample representative? It is also necessary to specify in detail the inclusion and exclusion criteria applied among the participants.
Response 3: Thank you for your suggestions. We have added more details related to this issue of period, sample representative and inclusion criteria.
Point 4: The material and methods section does not reflect different important ethical considerations either. It would be necessary to indicate if the authorization of an ethics committee was obtained for data collection. It also does not reflect whether the participation of the patients was voluntary or if they were offered a document with the information and informed consent.
Response 4: Many thanks, we have revised it.
Point 5: The results are clear and precise, including the necessary elements and offering greater rigor to the article. However, in the tables the acronyms included in them are not specified at the foot of the table.
Response 5: Thank you very much for this constructive feedback, we have added the acronyms at the foot of the table.
Point 6: The conclusions are specific and very appropriate. In addition, they point out the implication of the subject in clinical practice and today.
Response 6: Thank you for your positive response.
Point 7: The bibliographic references do not comply with the regulations required by the journal. I would recommend reviewing and adapting
Response 7: Thank you for your feedback. We have reviewed the journal guidelines.

Reviewer 2 Report
The authors here investigated the socio-economic and demographic factors that that influences knowledge about HIV/AIDS and attitude towards PLWH among 15–49-year-old Indonesian women by analysing publicly available data from the 2017 Indonesian Demographic and Health Survey (IDHS). Authors report that age, level of education, access to information, economic status, area/region of residence and social-economic independence (autonomy) are the main factors that’s determines level of knowledge about HIV/AIDS and attitude towards PLWHA (people living with HIV/AIDS). My comments follow.
MAJOR
1. In my opinion “….with knowledge and perception of HIV/AIDS…” or “with knowledge of HIV/AIDS and attitude toward people living with HIV/AIDS (PLWHA) are more grammatically correct.
2. The manuscript needs serious English language proofing, extensive rephrasing, major re-writing to improve comprehension and clarity. Providing specific will mean specifying the whole manuscript. E.g
· Introduction Line 3: HIV/AIDS infection rate and prevalence.
· Proof and clarify: “These PLWHA have physical, economic, social and psychological consequences as HIV infected patients were reportedly denied healthcare, poor quality treatment, refused when applied for jobs due to HIV/AIDS status, and forced early resignation from their employment [12]–[14].”
3. Introduction: “[11]. Several studies have demonstrated that people tend to have negative attitudes and report not buying vegetables from PLWHA status sellers.” Please include reference(s).
4. The discussion about how knowledge about HIV could increase awareness and reduce infection rate as well as how positive attitude towards PLWHA (reduced/no social stigmatization) could reduce death form HIV/AIDS by encouraging testing, early detection, ART uptake and social assimilation needs further expansion. Giving how interesting and depth of available research on the topic, authors have not appreciated this in the manuscript.
5. Method: Although authors have described and referenced the survey questions, a copy should be provided in this manuscript as a supplemental material.
6. The classification of knowledge and attitude into two categories based on ≥5 correct responses needs to be justified? For instance, does all the questions have presumably equal weight in measuring of knowledge and/or attitude?
7. Table 2 and other tables. P-values should be presented instead of the X2 scores.
8. In the "Multiple logistic regression analysis subsection of the result, the odds ratio <2 should be translated into "% more likely" to avoid repeating what is in the brackets. The OR more than 2 can be rounded up to whole number for easy interpretation e.g., 6.3 can be interpreted as 6 times more likely etc.
9. To reduce the size of the tables for easy and concise reading, I strongly recommend authors to only present statistically significant results. The complete tables could be presented as supplemental materials.
10. Also, the OR can be presented with the 95% CI [e.g., OR(95% CI)] and the significant level (p-values) and/or SEM can be presented as well.
MINOR
a. Abstract: Line 1, do you mean HIV/AIDS infection (not prevention)?
b. Correct to : These findings revealed that several demographical and social factors contribute to knowledge and attitude toward HIV/AIDS among women aged 15-49 years in Indonesia.
c. Rephrase as “knowledge and attitudes may affect behaviour toward HIV/AIDS [15].”
d. Please clarify this sentence : "To access information on the mass media of women from exposure to newspaper ≥1 per week as much as 1.32 times greater to have goof knowledge than never exposure to newspaper [OR= 1.32; 95% CI=1.04-1.67]. "
e. Clarify: "Women access to radio ≥1 per week and mobile phones were 1.31 [OR=1.31; 95% CI=1.11-1.56] and 1.26 [OR=1.26; 95% CI=1.12-1.44] times more like to have a good knowledge compared with never access to radio and mobile phone. Women ".
f. Remove : "Further information about associations is presented in Table 4. " since this table is already referenced earlier”.
g. Clarify: "reflecting personal awareness in escalating information on HIV/AIDS. ".
h. Consider rephrasing to: “Even though the study population were significantly knowledgeable about HIV/AIDS…”
i. Consider rephrasing to: “Our study highlights the need for public and/or specific group awareness about HIV/AIDS, as suggested in previous studies”.
j. Consider rephrasing to: “Our study indicates that a high knowledge about HIV/AIDS does not translate to more positive attitudes and revealed the critical social barrier toward PLWHA in Indonesia.”
k. Provide references for this : “lifetime, while attitude is complex processes within human perception that can changes constantly de-pending on specific situation.”
l. Please clarify : “Concerns about children’s status due to the ripple effect of stigmatization of the child and combination antiretroviral therapy raise the impact of PLWHA”.#
m. Clarify: These findings indicated the social, cultural, and religious beliefs that influence women’s identities, confirming other studies [40], [45].
n. Clarify: “In particular the cross-sectional study design, there was no causal relationships linked to this study.”
o. Clarify: “Our findings indicate that HIV/AIDS information and pro-grammes related to personal background, place, and mode of media improve high knowledge and positive attitude towards HIV/AIDS”.
p. Clarify: “Appropriate health education pro-grams indicated as the key to increasing comprehensive knowledge and considered would alter women’s attitudes and behaviors towards HIV/AIDS”.
Author Response
Response to Reviewer 2
Comments
Point 1: The authors here investigated the socio-economic and demographic factors that that influences knowledge about HIV/AIDS and attitude towards PLWH among 15–49-year-old Indonesian women by analysing publicly available data from the 2017 Indonesian Demographic and Health Survey (IDHS). Authors report that age, level of education, access to information, economic status, area/region of residence and social-economic independence (autonomy) are the main factors that’s determines level of knowledge about HIV/AIDS and attitude towards PLWHA (people living with HIV/AIDS). My comments follow.
In my opinion “….with knowledge and perception of HIV/AIDS…” or “with knowledge of HIV/AIDS and attitude toward people living with HIV/AIDS (PLWHA) are more grammatically correct.
Response 1: Thank you very much for this constructive feedback. We have changed “….with knowledge and attitude of HIV/AIDS….” because perception and attitude were different terms
Point 2: The manuscript needs serious English language proofing, extensive rephrasing, major re-writing to improve comprehension and clarity. Providing specific will mean specifying the whole manuscript. E.g
- Introduction Line 3: HIV/AIDS infection rate and prevalence.
- Proof and clarify: “These PLWHA have physical, economic, social and psychological consequences as HIV infected patients were reportedly denied healthcare, poor quality treatment, refused when applied for jobs due to HIV/AIDS status, and forced early resignation from their employment [12]–[14].”
Response 2:
- Thank you very much, we have added the details
- Many thanks, we have revised it on the manuscript section.
Point 3: Introduction: “[11]. Several studies have demonstrated that people tend to have negative attitudes and report not buying vegetables from PLWHA status sellers.” Please include reference(s).
Response 3: Thank you for your feedback and the resubmission has been updated with the references, including
[12] J. E. Ehiri, E. C. Anyanwu, E. Donath, I. Kanu, and P. E. Jolly, “AIDS-related stigma in sub-Saharan Africa: its contexts and potential intervention strategies.,” AIDS Public Policy J., vol. 20, no. 1–2, pp. 25–39, 2005.
[13] M. Dahlui et al., “HIV/AIDS related stigma and discrimination against PLWHA in Nigerian population,” PLoS One, vol. 10, no. 12, pp. 1–11, 2015, doi: 10.1371/journal.pone.0143749.
[14] A. Ben Moussa et al., “Determinants and effects or consequences of internal HIV-related stigma among people living with HIV in Morocco,” BMC Public Health, vol. 21, no. 1, pp. 1–11, 2021.
Point 4: The discussion about how knowledge about HIV could increase awareness and reduce infection rate as well as how positive attitude towards PLWHA (reduced/no social stigmatization) could reduce death form HIV/AIDS by encouraging testing, early detection, ART uptake and social assimilation needs further expansion. Giving how interesting and depth of available research on the topic, authors have not appreciated this in the manuscript.
Response 4: We have focused this study on the knowledge and attitude on HIV/AIDS and relate the finding with the sociodemographic factors.
Point 5: Method: Although authors have described and referenced the survey questions, a copy should be provided in this manuscript as a supplemental material.
Response 5: Thank you very much, the questionnaire is open and available to the public here https://dhsprogram.com/pubs/pdf/DHSQ8/DHS8_Womans_QRE_EN_8Apr2022_DHSQ8.pdf
Point 6: The classification of knowledge and attitude into two categories based on ≥5 correct responses needs to be justified? For instance, does all the questions have presumably equal weight in measuring of knowledge and/or attitude?
Response 6: Thank you very much, we have added the explanation on the variables section as follow: Knowledge was categorized as poor if respondents only answered correctly less than five, and categorized as good if respondents answered correctly ≥ 5. The attitude was divided into positive if respondents answered correctly ≥3 questions and negative if only answered 1-2 questions.
Point 7: Table 2 and other tables. P-values should be presented instead of the X2 scores.
Response 7: Thank you for your suggestion, We have revised as suggested: X2
Point 8: In the "Multiple logistic regression analysis subsection of the result, the odds ratio <2 should be translated into "% more likely" to avoid repeating what is in the brackets. The OR more than 2 can be rounded up to whole number for easy interpretation e.g., 6.3 can be interpreted as 6 times more likely etc.
Response 8: We appreciate the reviewer’s feedback on this matter. We have revised the sentence as follows:
The final multiple logistic regression models in Table 4 were adjusted for association across variables. Being 45-49 years old were 181% more likely to have a good knowledge of HIV/AIDS compared to women aged 15-19 years old [1.81(1.26-2.60)]. Women with higher education levels were 6 times more likely to have a good knowledge compared to women who have no education at all [6.32(3.59-11.11)]. Women with the richest wealth index were 167% more likely to have a good knowledge of HIV/AIDS compared to women with the poorest wealth index [1.67(1.32-2.10)]. Women who lived in urban areas and East of Indonesia Province were 137% [1.37(1.22-1.54)] and 128% [1.28(1.00-1.68)]more likely to have a good knowledge compared to rural area and middle of Indonesia Provinces respectively. Women who accessed information from the internet ≥1 per week were 174% more likely to have a good knowledge compared to women who never accessed to internet [1.74(1.50-2.03)]. To access information on the mass media of women from exposure to newspaper ≥1 per week as much as 132% greater to have goof knowledge than never exposure to newspaper [1.32(1.04-1.67)]. Women access to radio ≥1 per week and mobile phones were 131% [1.31(1.11-1.56)] and 126% [1.26(1.12-1.44)] more like to have a good knowledge compared with never access to radio and mobile phone. Women who have high autonomy were 123% more likely to have good knowledge compared with low autonomy [1.23(1.05-1.44)].
Point 9: To reduce the size of the tables for easy and concise reading, I strongly recommend authors to only present statistically significant results. The complete tables could be presented as supplemental materials.
Response 9: Many thanks for your suggestion, as we do not have many variables, we prefer to mention all variables on to table.
Point 10: Also, the OR can be presented with the 95% CI [e.g., OR(95% CI)] and the significant level (p-values) and/or SEM can be presented as wel
Response 10: Thank you for this important suggestion. This section has been revised following this suggestion.
Point 11: Abstract: Line 1, do you mean HIV/AIDS infection (not prevention)?
Response 11: Thank you for this important correction.
Point 12: Correct to : These findings revealed that several demographical and social factors contribute to knowledge and attitude toward HIV/AIDS among women aged 15-49 years in Indonesia.
Response 12: Thank you for your suggestion, We have revised as suggested
Point 13: Rephrase as “knowledge and attitudes may affect behaviour toward HIV/AIDS [15].”
Response 13: Thank you very much, we have changed the recommendation accordingly.
Point 14: Please clarify this sentence : "To access information on the mass media of women from exposure to newspaper ≥1 per week as much as 1.32 times greater to have goof knowledge than never exposure to newspaper [OR= 1.32; 95% CI=1.04-1.67]. "
Response 14: Thank you very much, we have revised this sentence clearly as follow:
Women who were exposed to newspaper ≥1 per week as much as 132% greater to have good knowledge than those never exposed to newspaper [1.32(1.04-1.67)].
Point 15: Clarify: "Women access to radio ≥1 per week and mobile phones were 1.31 [OR=1.31; 95% CI=1.11-1.56] and 1.26 [OR=1.26; 95% CI=1.12-1.44] times more like to have a good knowledge compared with never access to radio and mobile phone. Women ".
Response 15: Many thanks we have revised it.
Point 16: Remove : "Further information about associations is presented in Table 4. " since this table is already referenced earlier”.
Response 16: Thank you for your feedback. We have deleted it.
Point 17: Clarify: "reflecting personal awareness in escalating information on HIV/AIDS. ".
Response 17: Many thanks for this suggestion. We have revised it.
Point 18: Consider rephrasing to: “Even though the study population were significantly knowledgeable about HIV/AIDS…”
Response 18: Thank you very much, we have changed the recommendation accordingly.
Point 19: Consider rephrasing to: “Our study highlights the need for public and/or specific group awareness about HIV/AIDS, as suggested in previous studies”.
Response 19: Thank you very much, we have changed the recommendation accordingly.
Point 20: Consider rephrasing to: “Our study indicates that a high knowledge about HIV/AIDS does not translate to more positive attitudes and revealed the critical social barrier toward PLWHA in Indonesia.”
Response 20: Thank you very much, we have changed the recommendation accordingly.
Point 21: Provide references for this : “lifetime, while attitude is complex processes within human perception that can changes constantly de-pending on specific situation.”
Response 21: Thank you for your recommendations, the reference has been added.
- Auzenbergs et al., “Changing Perceptions: Talking about HIV and attitudes,” Positive Voices, 2018.
Point 22: Please clarify : “Concerns about children’s status due to the ripple effect of stigmatization of the child and combination antiretroviral therapy raise the impact of PLWHA”.#
Response 22: We have deleted irrelevant information, many thanks.
Point 23: Clarify: These findings indicated the social, cultural, and religious beliefs that influence women’s identities, confirming other studies [40], [45].
Response 23: We have deleted irrelevant information, many thanks.
Point 24: Clarify: “In particular the cross-sectional study design, there was no causal relationships linked to this study.”
Response 24: Many thanks, we have revised it.
Point 25: Clarify: “Our findings indicate that HIV/AIDS information and pro-grammes related to personal background, place, and mode of media improve high knowledge and positive attitude towards HIV/AIDS”.
Response 25: Many thanks, we have revised it.
Point 26: Clarify: “Appropriate health education pro-grams indicated as the key to increasing comprehensive knowledge and considered would alter women’s attitudes and behaviors towards HIV/AIDS”.
Response 26: Many thanks, we have revised it.
Round 2
Reviewer 2 Report
Most of my comments have been addressed.